# CERTIFIED ROBUSTNESS AGAINST PHYSICALLY-REALIZABLE PATCH ATTACKS VIA RANDOMIZED CROPPING

## ABSTRACT

This paper proposes a certifiable defense against adversarial patch attacks on image classification. Our approach classifies random crops from the original image independently and classifies the original image as the majority vote over predicted classes of the crops. This process minimizes changes to the training process, as only the crop classification model needs to be trained, and can be trained in a standard manner without explicit adversarial training. Leveraging the fact that a patch attack can only influence a certain number of pixels in the image, we derive certified robustness bounds for the classifier. Our method is particularly effective when realistic transformations are applied to the adversarial patch, such as affine transformations. Such transformations occur naturally when an adversarial patch is physically introduced in a scene. Our method improves upon the current state of the art in defending against patch attacks on CIFAR10 and ImageNet, both in terms of certified accuracy and inference time.

## 1 INTRODUCTION

Despite their incredible success in many computer vision tasks, deep neural networks are known to be sensitive to adversarial attacks; small perturbations to an input image can lead to large changes in the output. A wide range of defenses against adversarial attacks have been conducted in image classification, where the goal of the attacker is simply to change the predicted label(s) of an image (Kurakin et al., 2016a; Szegedy et al., 2013; Madry et al., 2017). But these defenses have typically considered a relatively unrealistic threat model that does not easily extend to the physical settings. In particular, these works have mainly considered the so-called $\ell_p$-norm threat model, where an attacker is allowed to perturb the intensity at all pixels of the input image by a small amount. In contrast, *adversarial patch* attacks are considered as physically-realizable alternatives, modeling scenarios where a small object is placed in the scene so as to alter or suppress classification results (Brown et al., 2017). Here, the attack is spatially compact, but can change the pixel value to any value within an allowable range.

This paper develops a practical and provably robust defense against patch attacks. Inspired by the randomized smoothing defense (Cohen et al., 2019; Levine & Feizi, 2019) for the $\ell_p$-norm threat model, our approach classifies randomly sampled sub-regions or crops of an image *independently* and outputs the majority vote across these crops as the class prediction of the input image. This approach has numerous benefits. First, given the size of adversarial patches, we can compute the probability of a sampled crop overlapping with the attacked region (patch), and use this probability to determine if the classification outcome of an image can be guaranteed (certified) to not be changed by any adversarial patch. Second, this approach is highly practical, as the crop classifier can be trained using standard architectures such as VGG (Simonyan & Zisserman, 2014) or ResNet (He et al., 2016) without the need for adversarial training. Indeed, random cropping is already a common data augmentation strategy for training machine learning models, and thus the method can be trained via standard techniques. This is different from most existing work on certifiable defenses against patch attacks (Levine & Feizi, 2020; Xiang et al., 2020; Chiang et al., 2020) which need extra computation for certification during training. Third, the proposed approach separates the training procedure from the patch threat model, thus making the method more robust against realistic settings of patch attacks, for example, patch transformations including rotation in x-y plane and aspect ratio

|  | CIFAR10 2.4% patch | | ImageNet 2.0% patch | |
|---|---|---|---|---|
|  | certification acc. (clean acc.) | time in ms | certification acc. (clean acc.) | time in ms |
| Proposed method | **47.5 (88.4)** | **0.7** | **12.2 (55.7 )** | **21.8** |
| De-rand. smoothing | 17.5 (83.9) | 17.5 | 3.2 (43.1) | 703.2 |
| PatchGuard: De-rand. smoothing | 18.2 (84.5) | 18.2 | 3.5 (43.6) | 734.5 |
| PatchGuard: Bagnets | 27.1 (82.6) | **0.7** | 9.6 (54.4) | 25.7 |

Table 1: Worst-case certified accuracy (%), clean accuracy(%), and certification time of the proposed method, De-randomized smoothing (Levine & Feizi, 2020), and PatchGuard (Xiang et al., 2020) with De-randomized smoothing and Bagnets as base structure. For each method, we list the *worst* certified accuracy under affine transformation of the patch. *Note that this is different from results in the original paper where patch transformations are not considered.*

changes. This is in sharp contrast to prior work that either have fixed strategies to exclude parts of the image or extract features from fixed parts of the image (Levine & Feizi, 2020; Xiang et al., 2020).

We summarize our main results on CIFAR10 and ImageNet in Table 1 in comparison with the current state of the art certifiable defense against patch attack (Xiang et al., 2020) with patch transformation. We report certified accuracy, which is the percentage of test images for which classification outcome equals to the ground truth label and is guaranteed to not change under patch attack. Our method is better in both speed and certified accuracy compared to De-randomized smoothing (Levine & Feizi, 2020) and PatchGuard (Xiang et al., 2020) under patch attack with possible affine transformations of the patch. In addition, our method outperforms these past approaches on ImageNet (though not on CIFAR10) in the setting where the patch aligns with coordinates of the image axes and does not undergo affine transformations as in Table 2, which was the setting considered in this past work.

We have made several contributions in this paper: first, we propose a defense against patch attack for image classification with certified robustness; second, the proposed method is fast in computing image certification and robust against patch transformation; third, the proposed method can be applied to any image classification model with only minimal changes to the training process.

## 2 BACKGROUND AND RELATED WORK

**Adversarial attacks** Adversarial attacks on image classification have been known for some time, with original work coming out of the field of robust optimization (Ben-Tal et al., 2009). Test-time attacks on ML models in general were studied in (Dalvi et al., 2004; Biggio et al., 2013), though the area gained momentum considerably when these methods were applied to deep learning systems to demonstrate that deep classifiers could be easily fooled by imperceptible changes to images (Szegedy et al., 2013; Goodfellow et al., 2014). In the following years many defenses against such attacks were proposed (Tramèr et al., 2017; Papernot et al., 2016), although most heuristic approaches were later found to be ineffective (Athalye et al., 2018). Amongst the defense strategies that have stood the test of time are: 1) adversarial training (Goodfellow et al., 2014; Kurakin et al., 2016b; Madry et al., 2017), now commonly carried out using a projected gradient descent based approach for synthesizing adversarial attacks and then incorporating them into the training; and 2) provably robust training (Wong & Kolter, 2017; Raghunathan et al., 2018). Our approach is more related to the randomized smoothing-based methods (Cohen et al., 2019; Levine & Feizi, 2019) in the latter direction, in which random points around the original input are sampled and classified, and the predicted class of the original input is declared as the aggregation of these outputs.

The majority of the attacks mentioned above focus on attacks with bounded $\ell_\infty$-norm, where attacks are permitted to modify any pixel in the image by (at most) some fixed amount (and are usually permitted to design a new adversarial perturbation for each new input image, though the so-called "universal" $\ell_\infty$ attacks have been studied as well (Moosavi-Dezfooli et al., 2017)). If we consider real-life attackers, however, this level of freedom on the attacker's side seems uncommon: attackers

would need to have access to the deep learning system at the pixel level, which if true, also means that the system has already been compromised, making it unclear what benefit such an attack would have. In contrast, *physically-realizable* attacks such that can be implemented in the real world have been primarily modeled by the so-called patch attacks (Brown et al., 2017; Eykholt et al., 2018), where a particular 'pattern' is designed to fool a deep learning system. We consider patch attacks as the threat model to defend against for it being physically realizable.

**Defense against patch attacks** There are also two threads of research in defending against patch attacks: 1) empirical defense strategies which show stronger empirical robustness but no analytical guarantees (McCoyd et al., 2020; Naseer et al., 2019; Hayes, 2018); most of these methods rely on the fact that a patch attack has to produce strong local features that are very different from its neighborhood to change classification outcome; by analyzing local features, one can detect patch location or smooth out such feature values to reduce effectiveness of the patch, and 2) certified defenses which provide analytical lower bounds of classification accuracy under patch attack.

**Certifiable defense against patch attacks** Relevant to our work is the defense which provides provable lower bounds of classification accuracy. In Chiang et al. (2020), interval bound propagation was used to provide certification, but the method is not applicable to commonly used image classification networks such as ResNet50 (He et al., 2016) or VGG16 (Simonyan & Zisserman, 2014) because the depth of these architectures renders the bound trivial, and thus cannot provide useful certification. Another line of research utilizes the fact that an adversarial patch can only have local influence on the image. To reduce such influence, De-randomized smoothing (Levine & Feizi, 2020) uses ablation to exhaustively block all possible patch locations while (Zhang et al., 2020) uses Bagnets (Brendel & Bethge, 2019) with limited receptive field to contain the number of affected features. Both methods then aggregate logits from ablated versions of images or local regions for final classification. PatchGuard (Xiang et al., 2020) adds additional detection of patch location and then sets features extracted from the detected locations to be zero. However all three methods require special image classifier structures – Levine & Feizi (2020) require three additional channels to express ablation, Zhang et al. (2020) is limited to Bagnets, and Xiang et al. (2020) uses either of the former structures. Our proposed method instead can use *any image classification architecture* and does not require any additional steps to compute certification during model training except cropping images. This makes our method much more practical and easier to deploy onto existing systems. We will compare our proposed method with De-randomized smoothing (DRS) and PatchGuard (PG) in the experiments and an overview of the two prior arts can be found in Appendix A.

## 3 PROVABLE PATCH DEFENSE WITH RANDOMIZED CROPPING

A physically-realizable adversarial patch attack can be found by solving the optimization problem:

$$\max_{\delta \in \Delta} \mathbf{E}_{(x,y) \sim D, t \sim T} \left[ \ell(h_\theta(A(x, \delta, t)), y) \right] \tag{1}$$

where $h_\theta : \mathcal{X} \to \mathcal{Y}$ denotes some (presumably deep-network-based) hypothesis function; $\theta$ denotes parameters of the model, $x \in \mathcal{X}$ denotes the original input to the network and $\delta \in \mathcal{X}$ the perturbation to the input; $\Delta \subseteq \mathcal{X}$ denotes the set of allowable perturbations; $y \in \mathcal{Y}$ denotes the true label; $\ell : \mathcal{Y} \times \mathcal{Y} \to \mathbb{R}$ denotes a loss function that measures the performance of the image classifier; and $\mathcal{T} : \mathcal{X} \to \mathcal{X}$ is the set of transformations that the perturbation $\delta$ might go through. The feasible set $\Delta$ denotes a simple allowable set of values for the patch, which typically would just be constrained to lie in valid RGB space. Patches *overwrite* a portion of the image with the patch perturbation itself, at a given location with a given set of transformations, such as scaling, rotation, and other transformations in $T$. We refer to this combination as the patch application function $A : \mathcal{X} \times \mathcal{X} \times \mathcal{T} \to \mathcal{X}$, where $A(x, \delta, t)$ denotes the application of patch $\delta$ to image $x$ with transformation $t$. Attack budget considered for patch attack is the percentage of area that the patch covers over the whole image. Throughout this paper, we consider the patch being one connected area bounded by a rectangle with size $p_i \times p_j$.

### 3.1 RANDOMIZED CROPPING DEFENSE

Although a patch attack can change pixel values to any arbitrary value, it can only influence the pixels within the patch itself. Therefore, networks extracting features with compact receptive fields

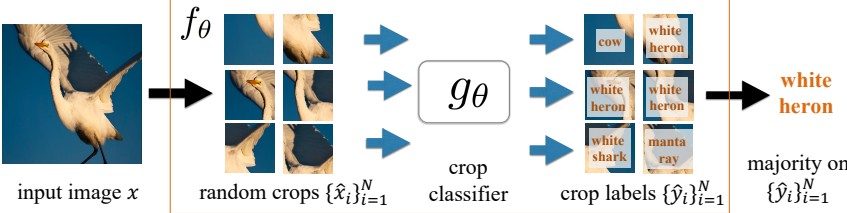

Figure 1: Forward pass of randomized crop defense

and aggregating such local features for final classification are more robust against patch attacks. An example of such methods is Bagnets (Brendel & Bethge, 2019), where 1x1 convolutional kernels are used to restrict the size of the receptive fields. Several existing techniques (Zhang et al., 2020; Xiang et al., 2020) utilize Bagnets for adversarial defense. However, the major drawback of Bagnets is that it extracts local features at fixed locations, therefore a patch may be placed in particular locations to influence more number of local features than other locations, thus reducing the worst-case robustness of Bagnets. In contrast to this architecture, we propose to extract local features at *random* locations by sampling random crops; this greatly reduces variations in the number of local features an adversarial patch can influence.

Based on above observations, we propose a randomized cropping approach as shown in Fig. 1: given a full-size test image $x$ as input, we first randomly select $n$ crops uniformly over all possible locations, where crop size $k_i \times k_j$ is smaller than image size $m_i \times m_j$ in both dimensions. Each of the sampled crops ($\widehat{x}_i$) then goes through the crop-based classifier $g_\theta$ and its predicted class ($\widehat{y}_i$) is obtained. The final classification of $x$ is the majority of $\{\widehat{y}_i\}_{i=1}^n$. The equivalent pseudo code of Fig. 1 is in Appendix B.

**Training randomized cropping classifier** The robust classifier $f_\theta$ has three components: (1) sampling $n$ crops with uniform distribution, (2) classifying crops with $g_\theta$, and (3) majority voting. Since both random sampling of the crops via a uniform distribution and majority voting have no trainable parameters, the set of trainable parameters of $f_\theta$ is the same as $g_\theta$. Therefore training the robust classifier $f_\theta$ only involves training the crop classifier $g_\theta$, which can be trained in standard ways with only data augmentation of randomly cropping the input $x$ as in Algorithm 1. Other variables are either pre-determined hyperparameters (crop size $k_i$, $k_j$), or can be adjusted at test time (number of crops $n$).

Since the training procedure of randomized cropping classifier $f_\theta$ only requires training the crop classifier $g_\theta$ without any assumption on attack parameters (size, shape, and location), nor does it depend on hand-crafted rules or thresholds, our method is robust against different patch shapes. We show supportive experimental results in Section 4 for this claim.

---

**Algorithm 1:** Training crop classifier for T epochs given crop size $k_i \times k_j$, image classifier $g_\theta$ with loss function $\ell$, and a dataset of $M$ images

---

**Input:** Full size image data $x$, labels $y$
**for** *t = 1, …, T* **do**
    **for** *m = 1, …, M* **do**
        Set $\widehat{x}$ as a crop of size $k_i \times k_j$ from image $x$ at a random location;
        $\theta = \theta - \nabla_\theta \ell(f_\theta(\widehat{x}, y)$ *//Update model weights with some optimizer, e.g. SGD*
    **end**
**end**

---

### 3.2 CERTIFIABLE ROBUSTNESS BOUND

Here we compute the probability that an image $x$ is certified robust, i.e., the classification outcome of $x$ cannot be changed by patch attack of given size. Given a clean image, we randomly sample $n$ crops $\{\widehat{x}_i\}_{i=1}^n$ and obtain the set of $n$ predicted classes $\{\widehat{y}_i\}_{i=1}^n$ of the crops. Let $n_1$ be the number crops that are predicted as the majority in $\{\widehat{y}_i\}_{i=1}^n$, and $n_2$ be the number crops that are predicted as

the second majority in $\{\widehat{y_i}\}_{i=1}^n$. Then if there exists a patch attack that can change the classification outcome of $x$, then the patch must overlap with at least $n_{2to1} = \frac{\lfloor n_1 - n_2 \rfloor}{2} + 1$ of the crops. On the other hand, if there are only less than $n_{2to1}$ crops that overlap with the patch, then $x$ is certified robust. Therefore, $p_c$ can be represented as the probability that, out of the $n$ sampled crops, there are less than $n_{2to1}$ crops that overlap with the adversarial patch.

The below derivation assumes that the random selection is uniform over all crop locations with replacement, i.e., the same crop can be selected again. Derivations for uniform sampling crops without replacement can be found in Appendix C.

Following the argument above, we first compute the probability that a single sampled crop $x_i$ overlaps the adversarial patch, and use this probability to compute $p_c$. This probability can be computed as the total number of crops overlapping with the patch (denoted by $n_{adv}$) divided by the number of all possible crops ($n_{all}$). Although a physically-realizable adversarial patch can be at any location in the image, to guarantee robustness, we consider the worst-case scenario where the patch is at the location that it can influence maximum number of crops (usually at the center of the image). When the patch is at the center of the image,

$$n_{adv} = \min(p_i + k_i - 1, m_i - k_i + 1) \times \min(p_j + k_j - 1, m_j - k_j + 1), \tag{2}$$
$$n_{all} = (m_i - k_i + 1) \times (m_j - k_j + 1), \text{ and} \tag{3}$$
$$p_a = \frac{n_{adv}}{n_{all}}. \tag{4}$$

After we sample $n$ crops $\{\widehat{x_i}\}_{i=1}^n$ from the original input $x$ and $\{\widehat{y_i}\}_{i=1}^n = \{g_\theta(\widehat{x_i})\}$ are the predicted classes of the crops. Suppose in $\{\widehat{y_i}\}_{i=1}^n$, $n_1$ of them are the majority class $y'$, and $n_2$ of them are the second-majority class $y''$. If an adversarial patch changes the majority class, the minimal number of crop-based predicted classes that it has to change will be

$$n_{2to1} = \frac{\lfloor n_1 - n_2 \rfloor}{2} + 1 \tag{5}$$

to make $y''$ the majority class.

The probability of certification $p_c$ is equal to the probability that out of $\{\widehat{x_i}\}_{i=1}^n$, at most $n_{2to1} - 1$ of them overlaps with the adversarial patch:

$$p_c = \sum_{i=0}^{n_{2to1}-1} C_i^n * p_a^i * (1 - p_a)^{(n-i)}, \tag{6}$$

where $C_i^n$ is the binomial coefficient ($n$ choose $i$). If $p_c$ is close to 1.0, the input image is certifiably robust under patch attack. In the experiments we select $p_c \geq 0.95$ for an image to be considered certified. Throughout this paper, we consider *certified accuracy*, meaning only data with predicted class $y'$ being the ground truth class will be considered for certification.

Note that because the crops are randomly sampled, $n_{2to1}$ and hence $p_c$ is an instance of a random distribution. However we argue that when $n$ is large enough, there $n_{2to1}$ and $p_c$ will have very small variance. Experiments to verify this point are in Appendix D.

The above computation of probability of certification in Eq. 6 again shows that $p_c$ is monotonically increasing with $n_{2to1}$. So to maximize number of certified robust images, the randomized cropping classifier should maximize $n_{2to1}$, which is equivalent to maximizing classification accuracy of $g_\theta$. This is another way to view that, to train the proposed robust classifier $f_\theta$, one only needs to train $g_\theta$.

## 4 EXPERIMENTAL RESULTS

**Datasets, models, and device.** We conduct experiments on two benchmark image classification datasets: CIFAR10 and ImageNet. For both datasets we evaluate over the whole evaluation set and report certified accuracy as percentage of images that are classified correctly and can be certified

| Patch Size | CIFAR10 | | | ImageNet | | |
|---|---|---|---|---|---|---|
| | ours | DRS | PG+DRS / Bagnets | ours | DRS smoothing | PG+DRS / Bagnets |
| 0.4% | 65.7 | 68.9 | **69.2** / 53.2 | 24.7 | 22.3 | **24.8** / 23.1 |
| 1.0% | 60.2 | 62.7 | **65.3** / 41.2 | **20.1** | 17.7 | 19.9 / 18.6 |
| 2.0% | 55.2 | 60.9 | **61.1** / 37.2 | **16.4** | 14.0 | 16.0 / 13.3 |
| 2.4% | 52.3 | 57.1 | **58.1** / 31.7 | **15.3** | 13.1 | 14.7 / 11.2 |
| 3.0% | 37.8 | 42.1 | **43.5** / 25.1 | **14.2** | 11.2 | 13.01 / 8.9 |

Table 2: Certified accuracy (%) over CIFAR10 and ImageNet of our method, De-randomized smoothing (DRS) (Levine & Feizi, 2020), and PatchGuard (PG)(Xiang et al., 2020). We consider images with $p_c \geq 0.95$ be certified.

with probability higher than 0.95. For CIFAR10, we use ResNet9 without the final pooling layer as the crop model, and for ImageNet we use ResNet34 as the crop model and resize and pad the image to 224x224. Experiments and timing are done on single Nvidia 2080 Ti GPU, and 16-core Intel i7-5960X CPU. For both models we use cyclic learning rate with $10^{-3}$ initial learning rate. The models are trained 30 epochs and training time is 14 GPU-hours and 328 GPU-hours for CIFAR10 and ImageNet, respectively.

we chose crop size of 10x10 (10% total area) for CIFAR and 80x80 (13% total area) for ImageNet; for these experiments we assume patch shape is *square with no rotation*. For each input image in CIFAR10, we randomly sample 128 crops and for each image in ImageNet we sample 256 crops.

**Positional encoding.** Because both CIFAR10 and ImageNet images have a certain region of interest which is usually in the center of the image, some parts of the images contain more information than others. For example, an image with label airplane usually has the airplane in the center, while the corners and edges have the background of sky or airport. Therefore, crops sampled at different position of the image contain different information. To represent such information, we add learnable positional encoding (Vaswani et al., 2017) to the first layer of our classifier.

**Metric.** Although the certification probability $p_c$ can be computed to any clean image despite of which data split this image is, in the experiment we follow the literature and report the certification accuracy, i.e., *percentage of test images that are certified and classified correctly*. This metric directly reflects how robust and accurate the classifier is – for example, the ideal classifier with 100% certified accuracy should classify all test images correctly and all test images' predicted class cannot be change by patch attack. We consider an image with $p_c \geq 0.95$ to be certified.

### 4.1 CLEAN ACCURACY AND INFERENCE TIME

Clean model accuracy and inference time per image compared with the above mentioned prior work are shown in Table 1. Note that clean model accuracy and inference time remain the same with and without affine transformation of the adversarial patch. In terms of inference time our method is faster under all cases. This is because although multiple forward passes are needed, the number of crop samples is smaller than equivalent sub-regions in Bagnets and much smaller than number of all band-smoothed images as in De-randomized smoothing (DRS). PatchGuard with Bagnets (PG-Bagnets) is slower than our method in ImageNet for the additional reason of having to go through all logits of all possible classes sequentially. In addition, both De-randomized smoothing (DRS) and PatchGuard with De-randomized smoothing (PG-DRS) as the base structure are *an order of magnitude* slower than our method because De-randomized smoothing passes the whole image through the classifier, which is much more expensive then passing crops (10% or 13% of the whole image) through the classifier.

|  | $p_c \geq 0.93$ | $p_c \geq 0.95$ | $p_c \geq 0.97$ | $p_c \geq 0.99$ |
|---|---|---|---|---|
| CIFAR10 2.4% | 52.8 | 52.3 | 52.0 | 49.6 |
| ImageNet 2.0% | 17.2 | 16.4 | 15.3 | 14.1 |

Table 3: Certified accuracy (%) with different thresholds for $p_c$ without patch transformation

| | CIFAR10 2.4% patch | | | ImageNet 2% patch | | |
|---|---|---|---|---|---|---|
| Transformation | ours | DRS | PG+DRS / Bagnets | ours | DRS smoothing | PG+DRS / Bagnets |
| AR 1:1 | 52.3 | 57.1 | 58.1 / 31.7 | 16.4 | 14.0 | 16.0 / 13.3 |
| AR 2.7:1 | 50.7 | 65.8 | 67.2 / 30.4 | 15.8 | 15.2 | 17.6 / 12.4 |
| AR 6:1 | 47.5 | 71.1 | 74.5 / 27.1 | 12.2 | 17.9 | 19.0 / 9.6 |
| AR 1:2.7 | 50.7 | 40.6 | 42.4 / 30.4 | 15.8 | 11.3 | 11.9 / 12.4 |
| AR 1:6 | 47.5 | 17.5 | 18.2 / 27.1 | 12.2 | 3.2 | 3.5 / 9.6 |
| Rotate 45° | 48.0 | 50.3 | 52.1 / 28.1 | 12.4 | 12.1 | 12.8 / 10.1 |
| Worst case | **47.5** | 17.5 | 18.2/27.1 | **12.2** | 3.2 | 3.5/9.6 |

Table 4: Certified accuracy (%) under patch rotation and different aspect ratio. We consider images with $p_c \geq 0.95$ be certified.

## 4.2 WITHOUT PATCH TRANSFORMATION

We first present the certification accuracy in Table 2 of our randomized cropping method along with De-randomized smoothing and PatchGuard on patch size ranging from $0.4\%$ to $3\%$. Position encoding used here is learnable position encoding with values up to 10% of normalized input value range. We mainly compare our results with two prior arts: De-randomized smoothing (DRS) (Levine & Feizi, 2020) and PatchGuard (PG) (Xiang et al., 2020). For all PatchGuard results there are two sets of numbers separated by slash: the first set of numbers use De-randomized smoothing(Levine & Feizi, 2020) as base structure and the second set of numbers use Bagnets (Brendel & Bethge, 2019). Methods with the highest certified accuracy are highlighted in bold. We can see that although our method is sub-optimal for CIFAR10, for ImageNet we have the highest certified accuracy except for patch size $0.4\%$. We also show the certification accuracy with different thresholds for $p_c$ Table 3.

As discussed in the previous subsection, our method is the fastest among the three methods. Although using Bagnets (PG-Bagnets) as base structure makes PatchGuard much faster than using De-randomized smoothing (PG-DRS), it pays the price of degrading certified accuracy. For example, certification accuracy degrades from 58.1% (PG-DRS) to 31.7% (PG-Bagnets) for CIFAR10 with 2.4% patch size while our randomized cropping defense is as fast as PatchGuard with Bagnets (PG-Bagnets) as show in Table 1 (inference time remains the same with or without patch transformation), but achieves much better (52.3%) certified accuracy. Moreover, on ImageNet our proposed method is the fastest and with the highest certified accuracy as well as clean accuracy.

Combining the certified accuracy on ImageNet in Table 2 and the short inference time in Table 1, our experiments show that the proposed randomized cropping defense is *practical in the aspects of fast certification and high certified accuracy* for the dataset that is closer to real-life pictures.

## 4.3 WITH PATCH TRANSFORMATION

As described in Section 3, a physically-realizable patch can be subject to certain transformations, such as scaling and rotation, before overwriting pixels in an image. These transformations can be used to mimic relative camera angle and distance to the scene in the real physical world. While it is reasonable to restrict the maximal area that an adversarial patch can influence in an image (since it corresponds to how big the physical patch is), one cannot assume such area would always align with coordinate axes of the image – even if the physical patch itself is square, when the scene is

captured with different camera angles, the patch on the image will be rotated (rotation in x-y plane) or stretched (rotation in depth). Therefore it is important that the patch defense can also provide guarantees when the patch is rotated or if the aspect ratio is varied, given the same patch size.

In Table 4 we compare the proposed randomized cropping defense with De-randomized smoothing (Levine & Feizi, 2020) and PatchGuard (Xiang et al., 2020) when a square patch is rotated 45 degrees and with aspect ratio (AR) 1:1, 6:1, 2.7:1, 1:2.7, 1:6, respectively. We highlight the highest certified accuracy under worst transformation for both datasets. Aspect ratio 1:1 without rotation is the same as Table 2 but listed here as reference. We chose 45 degrees because it is the worst case for a square patch, as the same number of pixels within the patch but occupying the largest rectangle area that aligns with coordinate axes of the image. Therefore rotation of 45 degrees provides a lower bound of robustness under patch rotation. For all transformations of the patch, we consider the same patch size.

We can clearly see that in Table 4, with the same patch size, transformation brings down the certified accuracy of all three competing methods, however, the proposed randomized cropping defense has the highest certified accuracy under worst-case patch transformation as listed in the last row of Table 4. This is because of our *random sampling* strategy that we have neither fixed the locations of sub-regions as in Bagnets, nor fixed smoothing strategy as in De-randomized smoothing.

On the other hand, the fixed column-smoothing strategy (ablating columns of images) in De-randomized smoothing (Levine & Feizi, 2020) which has the highest certifiable accuracy for square patches aligning coordinate axes of the image suffers particularly because as the patch gets shorter and longer (aspect ratio 1:3 and 1:6), the number of columns needed to ablate all the patch gets higher, and leaves fewer pixels of the image for classification. Also rotating the patch 45 degrees is equivalent to increasing the number of columns the patch occupies by 1.4 times, and hence significantly reduces certified accuracy for column-smoothing strategy. While there are other smoothing techniques in De-randomized smoothing (block smoothing), similar effects can be expected as some aspect ratio or rotation degree will cause significant degradation of certified accuracy. Same level of degradation is also seen in PatchGuard with De-randomized smoothing base structure (numbers in PatchGuard column before slash symbol).

## 4.4 DISCUSSIONS

**How to choose crop size?** Assuming that the number of crops $n$ is fixed, then in general larger crops leads to a better clean performance, as each crop contains more information when it covers more pixel area. Also with larger crops, the crop classification accuracy of $g_\theta$ would be better, indicating that $n_{2to1}$ in Eq. 5 could be larger and increases the probability of certification $p_c$. However, larger crops also means that the probability of overlapping the adversarial patch is higher (Eq. 7) which will decrease $p_c$. Therefore, for a given size of adversarial patch, there exists an optimal crop size which maximizes the certification probability.

To demonstrate the influence of crop size on certification accuracy and clean accuracy, clean and certified accuracy with or without positional encoding, with regards to different crop sizes are shown in Figure 2 for 2.4% patch on CIFAR10 and 2.0% patch on ImageNet – we use square crops and square patches aligning with coordinate axes of the image, i.e., no patch transformation. Comparing clean accuracy with and without positional encoding, we can see that although clean accuracy still increases as crop size increases when crop classifier includes positional encoding, but not as much as without positional encoding. Such results show positional encoding does provide extra information for crops sampled from different locations. On the other hand, comparing clean accuracy with certified accuracy, it is clear that the certified accuracy actually gets lower when crop size crosses some threshold as discussed above.

Note that similar experiments as in Figure 2 can be conducted for different sizes of adversarial patches to find optimal crop size. However we used fixed crop size in Table 1, 2, 4 to compare with DRS and PG to have a fair comparison, as these two approaches do not have tune-able ablation/kernel size.

**How to choose number of crops?** We show certified accuracy and inference time with different number of sampled crops for 2.4% patch on CIFAR10 and 2.0% patch on ImageNet. Certified accuracy of using all crops, i.e., selecting crops at all locations once without sampling, is plotted

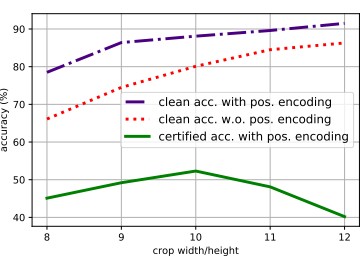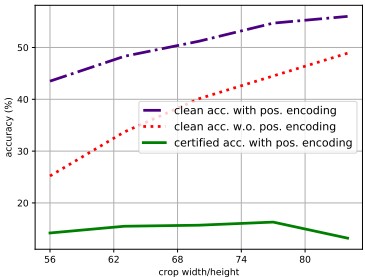

Figure 2: Clean and certified accuracy with different crop sizes. Left: CIFAR10 with 2.4% patch. Right: ImageNet with 2.0% patch

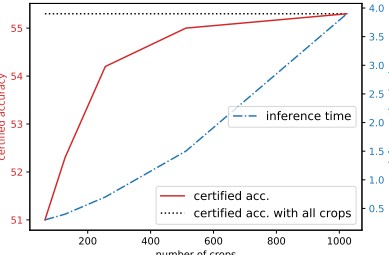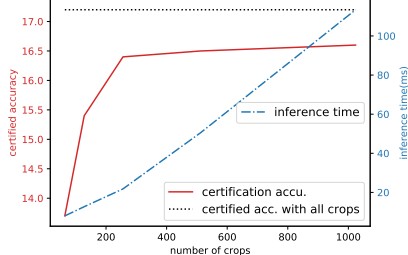

Figure 3: Certified accuracy and inference time(ms) with different number of sampled crops. Left: CIFAR10 with 2.4% patch. Right: ImageNet with 2.0% patch

with the black dotted line as reference. This line can be viewed as the upper bound of the proposed method. With more crops sampled, the certified accuracy increases but however inference time also increases close to linearly. Therefore we chose relatively small number of crop samples to balance between inference time and certification accuracy.

## 5 CONCLUSION

This paper proposes a new architecture for defense against physically-realizable patch attacks. The proposed approach decomposes an image into a random assortment of crops, each of which is processed by a classifier, and the majority across the crops is used as the classification outcome for the input image. This approach provides a significant advance in the form of improved certified accuracy, while maintaining a high clean accuracy when compared to prior art. We have shown that the proposed approach can be easily incorporated into standard training and test pipelines with minimal change to the underlying codebase. We believe this approach, by relying on a simple yet effective architectural change, provides a compelling alternative to traditional defense strategies and, in this sense, advances robust machine learning.

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

# A OVERVIEW OF DE-RANDOMIZED SMOOTHING (DRS) AND PATCHGUARD (PG)

De-randomized smoothing (Levine & Feizi, 2020) ablates all possible parts of the image and aggregates logits of the ablated images for certification. The method compares number of ablated images with high logit value of the majority predicted class with the number of ablated images with high logit value of the second majority predicted class. If the difference between the two is larger than two times possible number of "ablation blocks" affected by adversarial patch, then this image is certified robust against patch attack. The paper proposes two modes of ablation: block smoothing and band smoothing. Block smoothing ablates square blocks while band smoothing ablates one column of the image. To represent ablated regions, the image classifier accept three additional channels representing ablated pixels of the original RGB channels.

The main idea of PatchGuard (Xiang et al., 2020) is to detect possible patch location and mask these locations for downstream robust classification. To detect patch location, the method identify if there are any local regions that contribute abnormally strongly to a class. If there does exist such a region, it is considered as the potential location of a patch and the features extracted from the region is masked/discarded. Because such robust masking procedure can be combined with other robust classification approaches, the paper evaluates its robust masking with two classification models: De-randomized smoothing (PG-DRS) and BagsNet (PG-BN).

# B DETAILED CERTIFICATION PROCEDURE

We summarize the certification procedure for a single image in Algorithm 2. A more robust classifier should be able to certify and correctly classify a higher percentage of clean images in the test set.

---

**Algorithm 2:** Certify if patch attack can change the predicted class of an image $x$

---

**Input:** Full size image $x$, label $y$, crop classifier $g_\theta$
**for** $i = 1 \ldots n$ **do**
 | set $\widehat{x_i}$ as a $k_i \times k_j$ crop of $x$ at random location;
 | $\widehat{y_i} = g_\theta(\widehat{x_i})$ //*Predicted class of crop $\widehat{x_i}$*
**end**
$\widehat{y'}, \widehat{y''}$ = majority and second majority of $\{\widehat{y_i}\}_{i=1}^n$,
$n_1, n_2$ = number of crops classified as $\widehat{y'}, \widehat{y''}$, respectively;
**if** $\widehat{y'} \neq y$ **then**
 | return not certified
**else**
 | compute $p_c$ using Eq. 6, **if** $p_c$ is close to 1.0 **then** return certified, **else** return not certified
**end**

---

# C UNIFORM SAMPLING WITHOUT REPLACEMENT

In this section we derive certification probability and experimental results with uniform sampling without replacement. Let $n_{all}^i$ be the number of all possible location for the $i^{th}$ sampled crop, $n_{adv}$ be the number of crop locations that overlaps with the adversarial patch, and $p_a^i$ is the probability that the $i^{th}$ crop overlaps with the adversarial patch, then

$$n_{adv} = \min(p_i + k_i - 1, m_i - k_i + 1) \times \min(p_j + k_j - 1, m_j - k_j + 1), \tag{7}$$

$$n_{all}^i = (m_i - k_i + 1) \times (m_j - k_j + 1) - i, \text{ and} \tag{8}$$

$$p_a^i = min(1, \frac{n_{adv}}{n_{all}^i}). \tag{9}$$

The probability of the image being certified $p_c$ is then the probability of less than $n_{2to1}$ crops overlaps with the patch. The closed-form expression of $p_c$ is complicated yet not informative and hence omitted here. Comparing $p_a^i$ with the $p_a$ in Section 3, we can see that when sampling without replacement, the probability of sampling a crop that overlaps with the adversarial patch increases with

|  | CIFAR10 2.4% | | ImageNet 2.0% | |
| --- | --- | --- | --- | --- |
| num. of crops | clean (R/NR) | certified (R/NR) | clean (R/NR) | certified (R/NR) |
| 64 | 88.3/88.6 | 51.0/51.1 | 50.1/50.4 | 13.7/13.9 |
| 128 | 89.6/89.7 | 52.3/45.3 | 54.7/54.9 | 15.4/15.5 |
| 256 | 89.8/89.8 | 54.2/24.7 | 54.8/54.8 | 16.4/16.2 |
| 512 | 89.8/89.8 | 55.0/0.8 | 55.0/55.0 | 16.5/16.2 |

Table 5: Certified and clean accuracy (%) with (R) / without replacement (NR) over CIFAR10 and ImageNet vs number of crops without patch transformation. Patch size is 2.4% for CIFAR10 and 2.0% for ImageNet and crop size is 10x10 for CIFAR10 and 80x80 for ImageNet. We consider images with $p_c \geq 0.95$ to be certified.

|  | CIFAR10 2.4% | | ImageNet 2.0% | |
| --- | --- | --- | --- | --- |
| num. of crops | $n_{2to1}$ interval/variance | $p_c$ interval/variance | $n_{2to1}$ interval/variance | $p_c$ interval/variance |
| 64 | 1.3/0.6 | 1.9/0.9 | 2.1/0.8 | 3.0/1.0 |
| 128 | 0.8/0.4 | 0.8/0.5 | 1.4/0.5 | 1.5/0.7 |
| 256 | 0.5/0.2 | 0.5/0.3 | 0.9/0.3 | 0.8/0.4 |
| 512 | 0.2/0.1 | 0.2/0.1 | 0.6/0.1 | 0.5/0.1 |

Table 6: Averaged interval and variance of $n_{2to1}$ and $p_c$ (in $10^{-2}$) vs number of crops without patch transformation. Patch size is 2.4% for CIFAR10 and 2.0% for ImageNet and crop size is 10x10 for CIFAR10 and 80x80 for ImageNet.

the number of crops sampled, and hence decreases the probability of certification $p_c$. On the other hand, sampling without replacement enlarges the expected area that crops would cover, so the clean performance will be better than sampling with replacement.

We compare the certified and clean accuracy with and without replacement in Table 5. As number of crops increases, we can see that the gain of clean accuracy for sampling without replacement decreases because when number of crops increases, even sampling with replacement is likely to cover most of the pixels, and the gain is slightly more significant in ImageNet than CIFAR10. This may be because ImageNet images are in general more complex than CIFAR10 and having the crops covering more pixels over the image could help the overall classification. The certified accuracy for sampling without replacement gets worse than with replacement since the probability of sampling a crop increases more significantly as the number of crops sampled increases. This is particularly true for CIFAR10 – with image size 32x32 and patch size 10x10, the number of non-overlapping crops that does not overlap with the patch is only 484, out of 1024 all possible locations. This means when sampling 512 crops, there are at least 28 crops overlapping with the adversarial patch, which significantly decrease $p_c$.

## D EXPERIMENTS ON $p_c$ INTERVAL

In this section we show the interval of $n_{2to1}$ and $p_c$ with different number of crops sampled. We run the certification process over test set 200 times to obtain the interval and variance of $n_{2to1}$ and $p_c$ for each test image. Interval is defined as difference between the highest value and the lowest value. We ran this experiment over patch size 2.4% for CIFAR10 and 2.0% for ImageNet.

As shown in Table 6, with increasing number of crops, both interval and variance of $n_{2to1}$ and $p_c$ decrease, to negligible values.

