# OpenReview forum: "Certified robustness against physically-realizable patch attack via randomized cropping"
_ICLR.cc/2021/Conference — Reject_

### Official Review · AnonReviewer1 · 2020-10-23
**Surprisingly simple approach to an important problem**

**Rating:** 5
**Confidence:** 4

**Review:**

The authors propose a surprisingly simple statistical defense, that can certify the robustness of a classifier against patch attacks. This is achieved by randomly sampling small rectangular subregions of the perturbed image and classify these samples individually.

The paper is mostly well written. Certification methods, able to handle patch attacks, are of significant interest in real world applications like autonomous driving. The paper is charming because of the simplicity. However, some questions remain unanswered and the experimental evaluation is not very thorough, hence this paper is borderline.

Questions:
- Have the authors considered some kind of adversarial training?
- As far as i understood the paper, the certification method is probabilistic due to the probabilistic intersection of sampled subregions with the adversarial patch. Could this be circumvented by defining a fixed set of subframes that get classified? Then the number of overlapping subregions could be calculated precisely in advance and the certification would not be probabilistic.
- How does the right figure in Figure 2 look like? The current one seems to be the wrong one.
- In Table 1, Table 2 and Table 3, the authors provide certification rates. Could the authors also provide how many images of the not certified ones can be successfully attacked?
- When is $p_c$ as in Equation 6 considered to be close enough to 1, is it $0.95$ as in Section 4?
- How do the certification rates change for larger/smaller adversarial patches?

Comments:
- $C_i^N$ in Equation 6 should be defined.
- A vertical line between the CIFAR10 and the ImageNet results in Table 2 & 3 would improve readability.

I am willing to increase my score if my questions and concerns are addressed.

-------------------

After reading the authors response, i think that this work would benefit from an experimental comparison of their random method against a method relying on a deterministic crop selection, even if the certification rates of the deterministic method are inferior, because the certificates both methods are yielding are different: The resulting certificates from the deterministic method would be deterministic instead of probabilistic. Hence i will retain my score.

---

> ### Author Response · Authors · 2020-11-25
> **We revised the paper to improve readability, and ran experiments as suggested**
>
> Thank you for the positive and helpful review. We have addressed the comments in the updated manuscript. Below please find responses for the questions:
>
> While adversarial training is a strong empirical defense against adversarial attack, it does not provide robustness guarantees, i.e., given a clean/unperturbed image, an adversarially-trained classifier cannot certify if adversarial attack can or cannot change the predicted class on this image. On the other hand, our method can certify if the predicted class of a given image can or cannot be changed under patch attack. We have considered adversarially-training $f_{\theta}$, and it does improve empirical performance but decreases certification probability. This is because certification probability increases as the number of crops from \textit{clean} image being classified correctly increases, but adversarial training decreases performance on clean samples and consequently decreases certification probability.
>
> A deterministic crop selection covering the whole image was considered, but we found that random selection is easier to select the number of crops, while having comparable certified accuracy.
>
> We list the percentage of images that cannot be certified but can be successfully attacked by image-specific patches in the below table with the corresponding overall accuracy. We train our image-specific patch by training an adversarial patch which is placed at the center of the image for each image. We train each patch for 100 steps with step size 0.02. Please note that although some images are still correctly classified under image-specific patch attack, there can still exist a patch that can change its classification output. Due to time constraints, we only show CIFAR with patch size 2.4\% and ImageNet with patch size 2\% with no patch transformation.
>
> ___
>  CIFAR10 2.4\% patch                     ||  ImageNet 2\% patch
> ___
> | ours | DRS | PG+DRS / PG+Bagnets || ours | DRS  | PG+DRS / PG+Bagnets
> ___
> attack success rate      | 83.2  | 94.3 |92.1 / 87.3                        ||78.5 | 84.9  |83.2 / 79.3
> classification accuracy| 60.3  | 59.5 |61.4 / 40.4                        ||34.4 | 27.0  |30.1 / 31.2

---

### Official Review · AnonReviewer3 · 2020-10-26

**Rating:** 4
**Confidence:** 4

**Review:**

Update after the rebuttal:

I read other reviews and response from the authors and I decided to keep my score. Overall, I think paper still needs more
work. For example, incorporating details on confidence into the main paper and not just a section in the appendix is quite
important as otherwise the paper is misleading.

==================================================

-> Summary:
In this paper, authors propose a new certified defense against adversarial patches. They propose a model
which samples different patches from the original image, performs classification of these patches using neural network classifier and performs a majority vote to compute the output label. The guarantee is obtained by computing a probability that none of the sampled patches intersect adversarial patch.

-> Reasons for score:

I vote for rejecting this paper. The main issue I have is that their probabilistic guarantees lack confidence intervals and
therefore it is not clear how meaningful they are. Further issue is that the method works better than prior work only if
certain image transformations are applied, such as rotation. Yet, this critical part is not described formally: what is rotated, entire image or just a patch? Same holds for aspect ratio. Due to these problems, I cannot recommend acceptance for this paper.

-> Pros:

I think explanations of the method are easy to follow and writing is solid, with some parts that require more clarifications.

-> Cons:

The biggest problem I have with this paper is that authors do not consider confidence intervals for their guarantees. In particular, probability p_c computed in Equation 6 is tied to the particular n patches that were sampled which determines the value of n_{2to1}. This means that n_{2to1} is random variable and p_c that authors compute holds for just one sampled value of n_{2to1}. So for this bound to be useful, we need some form of confidence interval. For example, guarantee could be that with 99% probability p_c is interval [p_c_low, p_c_high]. And then, only if p_c_low is greater than some probability threshold (which is set to 95% in Section 4), we can report that this sample is certified. If I misunderstood something, it would be great if authors can clarify what kind of guarantee is provided.

Additionally, I find method used here relatively trivial and I think more technical contributions are necessary (e.g. computing confidence intervals above). In terms of empirical results, it seems that this method works better than prior work only in the case of additional image transformations and otherwise it performs worse or same. As this is the critical thing, I think authors should provide more explanations on how are these transformations applied, more formally instead of just describing it.


-> Questions:

- In Equation 6, summation is over i in the set {0, n_{2to1}}. Should this summation be over 0 <= i < n_{2to1} instead?
- Can you clarify what is the difference between the experiments in Table 1 and Table 2? If I understand correctly, the experiments in Table 1 have additional transformation that is applied besides the adversarial patch?
- Can you write mathematical formulation of transformations in Section 4.2? I am not sure whether these transformations are applied over the entire image or only over the patch, so it would be good to write more formally what exactly is the transformation.

---

> ### Author Response · Authors · 2020-11-24
> **We added confidence level experiment in Appendix D**
>
> Thank you for the helpful review. We have made corrections to Eq. 6 and added experiments and discussion of confidence level in Appendix D.
>
> Certified accuracy listed in Table 1. (and Table 3) are when the adversarial path undergoes affine transformation, while adversarial patches in Table 2 are all square with edges align with image coordinate axes. For example, a 2.4\% patch for CIFAR10 is always a 5x5  square and its edges align with image coordinates, i.e., edges of the patch are parallel to image edges for Table 1; a 2.4\% patch for CIFAR10 can be a 2x12 or 3x8 rectangle for Table 2.

---

### Official Review · AnonReviewer2 · 2020-10-28
**The paper proposed a simple way to defense adversarial patch attacks.**

**Rating:** 5
**Confidence:** 4

**Review:**

The method can be basically summarized as a majority voting of crops of an image. Moreover, a new certification of the proposed method is introduced, not similar to the conventional adversarial robustness certification on perturbation under $\ell_p$ ball , the method using simple geometry and probability problem to certify the results instead of any relaxation of the neural networks.
However, the paper is poorly written, and many experimental setups are missing.

Update after rebuttal:
After reading the rebuttal, I don't think my questions are addressed very well, especially about the confidence probability, $p_c$. The certification is defined as a guaranteed yes/no problem but the $p_c$ will relax the certification to a probabilistic problem. Also, PatchGuard with patch transformation is out of scope for the original paper, so I think the experimental results in Section 4.1 and 4.2 are more like a fair comparison. However,  refer to the results in table 3, the proposed method yields worse performance than PG-DRS although the computational cost is saved. Hence, I will keep my rating.

Pros:

+The proposed method is quite general and intuitive.

+Using a simple geometry and probability model, the proposed method can certify the robustness of the adversarial patch attack in a very efficient way.

Cons:

-What's the training time of the crop classifier $g$ compare with other baselines?

-The whole system highly depends on the test accuracy of $g$. What's the result if no attack involved in?

-From the experimental results, the proposed method has margin improvement compare with PG+DRS, and the basic idea of the certification algorithm also follows PG+DRS. So, the contribution is not enough for ICLR.

-The hyperparameter $p_c$ is set as 0.95 in the experiment. However, the certification needs guaranteed results. So, if the $p_c = 0.95$, is that means there is a 5% possibility that this image can be attacked? Although this may not change the results so much, the ablation study of this parameter setting should be performed to make the comparison fairly.

-Figure 2 (right) seems to use the wrong plot.

---

> ### Author Response · Authors · 2020-11-24
> **We have added ablation study of $p_c$ and revised the manuscript for improved readability**
>
> Thank you for the helpful review. We've added training time to the first paragraph of Section 4. Clean accuracy are listed in Table 1 and a discussion on clean accuracy is added to Section 4.1. The ablation study of $p_c$ is added to Table 3.
>
> We respectfully disagree with the reviewer that our method has only marginal improvement over PG+DRS -- without patch transformation, we achieve comparable certification accuracy as PG+DRS while the inference time is more than a magnitude faster; with patch transformation, our worst-case certification accuracy is 4 times better than PG+DRS. This comes from our basic idea of using crops instead of ablating regions of an image.

---

### Official Review · AnonReviewer4 · 2020-11-02
**An elegant defense. But text and discussion must be improved.**

**Rating:** 5
**Confidence:** 4

**Review:**

### Summary

The paper proposes a robust neural network architecture to defend against
adversarial patch attacks --where the attacker can freely modify a small patch
of size $s_{patch}$ of the input images-- by training the net on random
crops of a pre-defined size $s_{crop}$ (in general $\neq s_{patch}$), and, at
test time, use majority voting over several such random crops for every image.
They show that, as long as $s_{patch}$ is small ($\leq 2.4\%$ of total image),
these architectures achieve usual and **certified** robust accuracies that are
competitive with the SOTA among the techniques and architectures that were
tailored to certifiably defend against such attacks.


### Overall evaluation

The defense is elegant for its simplicity and seems efficient, both in terms of
performance for small attack patches (which, arguably, is the most interesting
case), and in terms of computational speed/complexity (for small enough sample
size of crops). But the paper is still unclear, vague and/or too
approximate in some parts (see points below). It seems to have been written in
a rush (Fig.2 on the right is obviously not the one intended) and some
additional experiments and/or discussions would be useful to complement the
evaluation of the current method (see point 8. below). So, overall, I don't
recommend publication in the current state but am inclined to reconsider when I
will see appropriate changes in the paper.


### Detailed remarks/questions

1. Since you repeatedly refer to de-randomized smoothing and PatchGuard in the
   text (and their combination), I suggest adding a short self-contained
   description (in appendix with a reference) at the beginning (or in appendix
   with a reference at the beginning) for the non-informed readers.
2. I suggest to clearly mention that what is certified is the robust **test**
   accuracy, not the distributional robust accuracy.
3. I think that you are sampling the crops **uniformly** at random **with**
   replacement, but I don't recall reading this explicitly in the text. Anyway,
   it would be good to compare both approaches (with and without replacement),
   at least in appendix.
4. The description of Table 1 is in Sec.4.1 with title "Without patch
   transformation", yet the caption says (and Table 2 confirms) that it shows
   results **with** patch transformations. Which one is right? And if it's with
   patch transformation, please add the same table for the setting without patch
   transformation (at least to appendix).
5. Eq. 6: capital $N$ in $C^N_i$ should be $n$.
6. Please do an additional pass of proof-reading
7. Sec 3.2: the description of how to compute the certifiable robustness bound
   could be **significantly** improved and seems to be written in a big rush. In
   particular, you never explicitly say which formula is used to compute the
   robust bounds.
8. _Crop size vs patch size_: The trade-off between crop size and attack patch
   size is discussed in a rush at the end of the experiments (Sec. 4.2): it
   deserves attention and an intuition explanation of the trade-off much earlier
   in the paper, f.ex. when explaining the overall attack. Also, the paper could
   benefit a lot from a finer analysis of the dependence of the optimal crop-size
   on the attack's patch size (and probably on the typical size of "relevant
   information" in the images). This could be either empirical (studying the
   optimal crop-size as a function of the patch size) or more theoretical (with an
   image model, or by using the average size of relevant objects in the image).
   Also, it could shed more light on why your attack works better on ImageNet than
   on CIFAR10 (probably because the typical size of the relevant parts of the
   images are different in both datasets).
9. _Sec. 3.1 §Training randomized cropping classifier_: the sentence "the only
   trainable part of the randomized cropping classifier is the crop
   classifier" deserves a proper Proposition/Theorem (with precise and explicit
   hypotheses, s.a., f.ex. the assumption that crops are sampled **uniformly**
   at random with replacement) and a proper, well-delimited proof.

------------------------------------

Update after rebuttal
-----------------------------

The updated version is clearly better than the first one. However, the concerns
of the other reviewers regarding the randomness of $p_c$ (and therefore of the
certification method) have convinced me that there are, indeed, some further
clarifications and discussions needed prior to the publication, which is why I
will keep my initial recommendation.

To be more precise, the root issue here seems to be that the proposed
classifier is not deterministic, which means that the standard definitions of
adversarial examples and adversarial accuracy do not apply and therefore, that
the problem that you try to solve is unclear and/or not well defined.  In
particular: what is it that gets certified?  what does it mean to get
certified? and, more generally, is the word "certified" really appropriate in
this context?

However, whether an analysis of the distribution of $n_{2to1}$ and $p_c$ will
be needed (as asked by other reviewers) might depend on how the authors will
define adversarial vulnerability in the random setting, and what they try to
certify. Let me explain what I mean.

A reasonable start might be to define adversarial risk as
$$
    E_{(x,y)} E_{\phi} \mathcal{L}(\phi(x), y) \ ,  \tag{1}
$$
which is the usual definition, but with an additional expectation over the
variability of the classifier $\phi$.  Adversarial accuracy would then be the
adversarial risk for the 0-1-loss $\mathcal{L}_{0-1}$. Then the authors could, f.ex., set as goal
to construct a (provably) unbiased estimate of this quantity.

The advantage of such a method is that one doesn't forget the fact that, what
we actually want to certify is this "distributional" robustness (i.e. where
expectation is taken over the true underlying, unknown distribution), not the
robustness on the test set. Even methods that have a non-random certification
process (so-called "provable robustness guarantees") will never be able to
certify this quantity: they'll only deliver certificates on test example. The
"certified robustness on the test set" that they yield is also just a random
variable which we hope "generalizes to" (1). Reviewers almost never ask authors
to analyze/certify this generalization gap. Similarly, here, one could see the
randomness over $n_{2to1}$ and $p_c$ as just another source of randomness
contributing to the variability of the generalization gap, in which case, maybe
no rigorous analysis could be acceptable, as long as it is clear what the
authors want to certify (unbiasedness of the estimator, f.ex.). Therefore,
whether this source of randomness could or should be explicitly captured/used
by the authors' method is, I think, a question of how the authors frame the
problem and their goal.

### Minor points:

- even the revised version still contains quite a few grammatical errors,
  especially in the new/re-worked sections, where many articles ("the", "a")
  are missing.

- End of p.5, "to maximize number of certified robust images, the randomized
  cropping classifier should maximize n2to1, which is equivalent to maximizing
  classification accuracy of $g_\theta$": not sure about this equivalence.
  Maximizing the classification accuracy is equivalent to maximizing n1, not
  necessarily n1-n2.

- are DRS and PG also probabilistic certifications (i.e. certifying robustness
  with some probability, f.ex. pc>.95)? This should be clearly said in the text
  and the captions, especially since it would make the comparison a bit unfair
  if their certification were 100% sure. (This issue is obviously related to my
  previous major remark on randomness.)

- the name "worst-case certified accuracy" in the caption of Table 1 is very
  unclear at that point. It becomes clear in Sec. 4.3, but you refer to Table 1
  in Sec. 4.1 already. So this term should be clearly explained in the caption,
  or there should be a clear reference to the relevant part in the text.

- don't always re-cite Levine&Feizi and Xiang et al. every time you mention
  de-randomized smoothing and patch guard. Cite them the first time, and then
  say that you'll refer to de-randomized smoothing and patch guard as DRS and
  PG in the rest of the text.

- in the conclusion: "This paper proposes a new architecture for defense
  against" -> "This paper proposes a new defense against". (You are not really
  proposing a new architecture.)

---

> ### Author Response · Authors · 2020-11-24
> **We've added suggested analysis and extra experiments in Appendix, and revised the manuscript for improved readability**
>
> Thank you for the thorough and helpful review. We've added a brief overview of De-randomized smoothing and PatchGuard in Appendix A, pseudo code explaining how certification probability is computed in Appendix B, and derivations and experimental results for uniform sampling without replacement in Appendix C. We've also expanded discussion on how certification rate is reported at test time in the "metric" paragraph in Section 4 and the effect of crop size vs patch size in Section 4.4. We analyze all three components of $f_{\theta}$ in the "Training randomized cropping classifier" paragraph of Section 3.1 and show that $f_{\theta}$ has the same set of trainable parameters as $g_{\theta}$. We had several passes of revising the manuscript to improve the readability.

---

### Decision · Program_Chairs · 2021-01-07
**Final Decision**

**Decision:**

Reject

**Comment:**

The paper provides a simple prediction procedure to defend against (rectangular) patch attacks, and also a method to obtain some random estimates of the certified robustness of the method. The simplicity of the method is certainly appreciated. On the other hand, there are a number of issues preventing the acceptance of this paper. The main problem is that the paper deals with a randomized predictor, yet the certification guarantee developed for deterministic predictors is applied. This leads to several problems, starting from the target being undefined to unfair comparisons. While the authors made an attempt to address this in the rebuttal, more work is needed to properly settle this issue.